# Evaluation of a health promotion intervention associated with birthing centres in rural Nepal

**Preeti Mahato**[1]\*, **Edwin van Teijlingen**[1,2,3], **Padam Simkhada**[2,3,4], **Catherine Angell**[1], **Vanora Hundley**[1]

**1** Centre for Midwifery, Maternal & Perinatal Health, Bournemouth University, Bournemouth, United Kingdom, **2** Manmohan Memorial Institute of Health Sciences, Tribhuvan University, Kirtipur, Nepal, **3** Nobel College, Pokhara University, Pokhara, Nepal, **4** School of Human and Health Sciences, University of Huddersfield, Huddersfield, United Kingdom

☯ These authors contributed equally to this work.
\* pmahato@bournemouth.ac.uk

**Data Availability Statement:** The data can be found in Bournemouth Online Research Data Repository (BORDaR) and the DOI is https://doi.org/10.18746/bmth.data.00000120.

## Abstract

### Introduction

Birthing centres (BC) in Nepal are mostly situated in rural areas and provide care for women without complications. However, they are often bypassed by women and their role in providing good quality maternity services is overlooked. This study evaluated an intervention to increase access and utilisation of perinatal care facilities in community settings.

### Methods

This longitudinal cross-sectional study was conducted over five years in four villages in Nepal and included two BCs. An intervention was conducted in 2014–2016 that involved supporting the BCs and conducting a health promotion programme with local women. Population-based multi-stage sampling of women of reproductive age with a child below 24 months of age was undertaken. Household surveys were conducted (2012 and 2017) employing trained enumerators and using a structured validated questionnaire. The collected data were entered into SPSS and analysed comparing pre- and post-intervention surveys.

### Results

The intervention was associated with an increase in uptake in facility birth, with an increase in utilisation of perinatal services available from BCs. The post-intervention survey provided evidence that women were more likely to give birth at primary care facilities (OR 5.60, p-value <0.001) than prior to the intervention. Similarly, the likelihood of giving birth at a health facility increased if decision for birthplace was made jointly by women and family members for primary care facilities (OR 1.76, p-value 0.023) and hospitals/tertiary care facilities (OR 1.78, p-value 0.020. If women had less than four ANC visits, then they were less likely to give birth at primary care facilities (OR 0.39, p-value <0.001) or hospitals/tertiary care facilities (OR 0.63, p-value 0.014). Finally, women were less likely to give birth at primary care facilities if they had only primary level of education (OR 0.49, p-value 0.014).

**Funding:** This was a PhD study (PM) which was supported financially through the fund obtained from Open Society Foundation (IN2016-30452). The support was mostly available for the fieldwork done in Nepal.

**Competing interests:** The authors have declared that no competing interests exist.

## Conclusion

BCs have the potential to increase the births at health facilities and decrease home births if their services are promoted by the local health promoters. In addition, socio-economic factors including women's education, the level of women's autonomy and having four or more ANC visits affect the utilisation of perinatal services at the health facility.

## Introduction

Proven interventions such as skilled birth attendance provided through a continuum of care that links households and communities with health systems, could prevent thousands of maternal and neonatal deaths in the world [1]. Skilled birth attendance requires the presence of a skilled attendant along with enabling environment including adequate supplies, equipment and infrastructure plus an efficient and effective communication and referral system [2]. Skilled birth attendants (SBAs) are competent maternal and newborn health professionals who are educated, trained and regulated to national and international standards [3]. The proportion of births attended by skilled health personnel are part of indicator 3.1.2 of the Sustainable Development Goals (SDGs) [4]. Measuring and monitoring of SBA remains a challenge because of the wide variety of definitions used. One study found uncertainty and diversity of reported qualifications and competency of SBAs between low- and middle- income countries and a need for improved coverage measurement and monitoring of SBAs [5]. Although there are many deaths caused by complications of pregnancy [6], evidence shows that the majority of women in low income countries, including Nepal, still give birth at home or in community settings without SBAs and in the absence of facility-based services [1, 7, 8]. In Nepal, one woman dies every eight hours due to complications in childbirth and 38 newborns die every day largely from preventable causes [9].

Nepal has seen a significant drop in its maternal mortality ratio as a result of an increase in the number of facility births and women being assisted by a SBA [10], however the remote and rural nature of the country means that many women still give birth at home without trained support. The solution may be the promotion of birthing centres (BCs) located closer to where women live.

Birthing Centres are a component of health system at local level designed to provide care for women experiencing a natural vaginal birth without complications. BCs provide a midwifery-led model of care where SBAs provide maternity services in a community or hospital setting normally to healthy women with uncomplicated or low risk pregnancies [11]. In Nepal, essential obstetric care (EOC) services are available at three levels of care: i) basic obstetric care available at Health Posts (HPs) including stabilising patients with obstetric first aid, making an appropriate referral and arranging transport; ii) basic emergency, obstetric and neonatal care (BEmONC) available at Primary Health Care Centres (PHCCs) to prevent and treat haemorrhage, treat puerperal sepsis, eclampsia, infection and manage prolonged labour; and iii) comprehensive emergency obstetrics and neonatal care (CEmONC) available at hospitals (central, provincial and district) to manage all the above plus caesarean section, anesthesia and blood transfusion [12]. In Nepal, a BC is usually established in rural areas at HPs and PHCCs and the number of BCs reported in the year prior to this study (2015/16) was 1,755 [13]. With a cadre of adequately trained SBAs in BCs it has been possible to provide basic essential obstetric care services in an effective way [14].

In most BCs, Auxiliary Nurse Midwives (ANMs) provide much of the primary care maternity services in Nepal. ANMs have 18 months of pre-service training in nursing and midwifery after ten years of schooling. They are trained to assist normal births, identify complications (and refer women to more specialist care) and offer health promotion. They are mostly deployed in BCs in rural Nepal where there is a lack of proper health facilities [15] but some are also deployed in urban hospitals.

BCs in Nepal are often bypassed in the hope of getting better quality services offered by hospitals [16, 17]. In this context, the role of BCs in providing good quality maternity services has often been overlooked in the case of Nepal [18]. Since BCs in Nepal are mostly present in rural areas, it is important that it provides quality services in order to increase its utilisation. Community-based health promotion interventions, which mobilise the community through facilitated participatory learning to improve access to, and use of, skilled care during pregnancy, childbirth and after birth, are highly recommended [19, 20]. Previous research in rural Nepal found that women's groups, based on participatory learning and action, led to improved maternal and newborn survival [21]. Thus, an intervention supporting BCs and providing community-based health promotion messages to community women [18] would appear to be an appropriate mechanism to improve maternity care.

Improving maternal health and outcomes requires a complex intervention. Increasing only the number of SBAs at BCs would not increase the uptake of services available at BCs; other enabling factors such as effective training, appropriate infrastructure, on-going professional development for staff, sufficient supplies and equipment, support from community health workers and effective referral mechanism are equally important [22]. This paper evaluated the effects of an intervention consisting of supporting BCs and community-based health promotion programme on increasing access and utilisation of perinatal care facilities in community settings.

## Materials and methods

The study area consisted of four village development committees (VDCs) in Nawalparasi district in southern Nepal that included two BCs where an intervention was conducted. A VDC used to be the smallest administrative unit at local level, but was dissolved in March 2017, just after conducting this survey [23]. The intervention, that was conducted by a local non-governmental organisation (NGO) during 2014–16, involved supporting two BCs and conducting a health promotion programme with local women in four VDCs. These two BCs started functioning in the year 2015 and 2016. The support included refurbishing the health facilities' infrastructure, providing equipment required for normal delivery, training all the ANMs at these two BCs and appointing two local ANMs as health promoters. An additional fund equivalent to US $50 was also provided monthly to the BCs for purchasing necessary instruments and materials. The health promotion programme consisted of training local health promoters, who then trained Female Health Community Volunteers (FCHVs). Prior to this intervention, no specific health promotion intervention existed apart from the basic health promotion role that all FCHVs have. The health promoters conducted meetings with mothers-in-laws as a strategy for creating demand for utilisation of the BCs [24]. For example, in 2016, there were 157 mothers-in-law meetings and 334 women's group meetings. The health promoters and FCHVs also met mother groups on a monthly basis and discussed various issues related to women's health through a curriculum covering content on ANC/PNC, baby feeding, sanitation and hygiene. The classes were informal and participatory and lasted one to two hours [24].

A longitudinal (repeated cross-sectional) study was conducted over a period of five years. The pre-intervention survey was conducted in the year 2012, the intervention was carried out

in 2014–2016 by the local NGO and we conducted a post-intervention survey in 2017 as part of this study. The data from pre-intervention survey were received from the NGO which conducted this survey. Being a repeated cross-sectional longitudinal study, the subjects were largely different from each other on each sampling occasion, although the area of study was the same [25]. The effects of the intervention were measured in this study.

Population-based multi-stage sampling (Fig 1) of women of reproductive age and having a child below 24 months of age was undertaken for both pre-intervention and post-intervention survey. Being a household survey, the eligible participants from each household of 29 selected wards of four VDCs who agreed to take part were approached by trained enumerators and a structured validated questionnaire [26] was completed. In order to get a spread on poorer and slightly better off wards as well as those closer to the BCs and those further away, 29 out of 36 wards were included in the study (Fig 1). The questionnaire used for the pre-intervention survey was adapted from the Nepal Demographic and Health Survey, the Water and Sanitation Survey and wider literature. For the post-intervention survey, the questionnaire was modified slightly based on experience of conducting the pre-intervention survey and removing some unnecessary questions related to socio-demographic characteristics.

The pre-intervention survey was conducted by a local NGO. This primary data from pre-intervention survey was made available to the first author who conducted a secondary analysis [27]. Post-intervention data were collected and a trainee, an undergraduate public health student, helped enter the data. All eight female data enumerators had a degree level qualification in a health subject. They received training (two days) from the first author.

A ten percent of sample was cross checked by the first author identifying a small number of discrepancies in data entry and these were corrected by the first author who then used this to supervise the trainee for future data entry. The pre- and post-intervention surveys were compared to identify any changes that might have occurred due to the intervention and also to determine the factors affecting place of birth. The outcome variables of the intervention were birth at BCs (primary outcome), number of ANC, women's decision making about place of

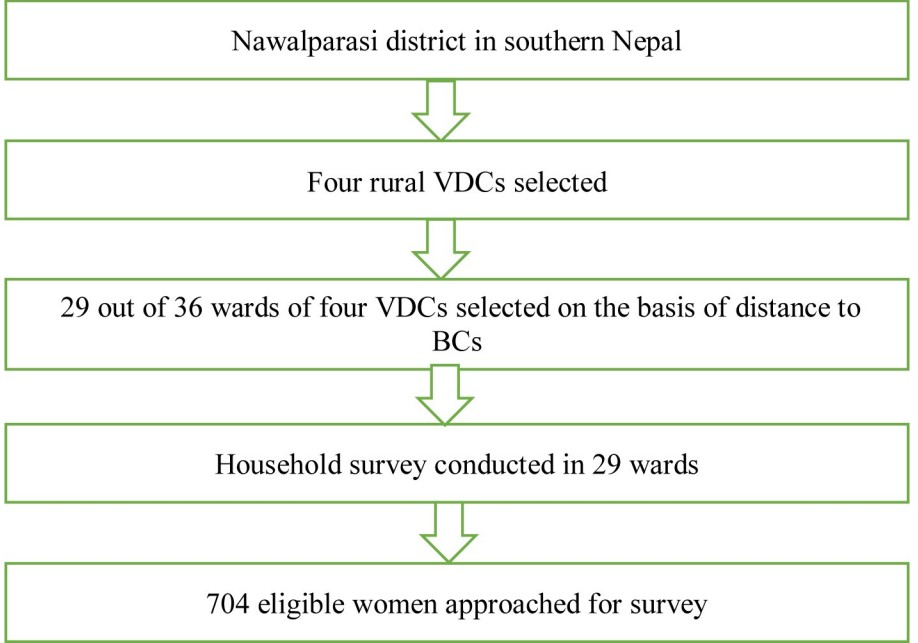

**Fig 1. Population-based multistage sampling.**

birth and satisfaction with childbirth services (secondary outcome). The primary outcome changed to various places of birth–home, primary care and tertiary care facilities because the descriptive findings of survey (pre-intervention) showed the data related to place of birth consisted of three categories. Descriptive analysis, cross-tabulation (chi-square), and multinomial regression analysis were conducted. To establish the strength of association between variables and where the association lies exactly, cross tabulation of the intervention with other significant variables was conducted. For the multinomial regression analysis both the pre- and post-intervention surveys were combined and the effect of each variable on birthplace was measured.

Ethical approval for this study was granted by University Research Ethics Committee (UREC) at Bournemouth University (Reference 8710) and Nepal Health Research Council (NHRC). In addition, informed consent was taken from the participants, either in written or verbal form. The consent process was clearly described in Nepali and was also explained verbally to all participants by the trained enumerators. Those participants who were able to read and write provided their consent by signing the participant information sheet and those who were not able to read and write provided their consent in verbal form. Verbal consent was witnessed and documented by the trained enumerators. Participants were made aware that taking part in the survey was voluntary and that information they provided would remain anonymous. The data were stored in a password-protected computer.

## Results

Among 704 women approached for the post-intervention survey, one did not take part in the survey and four were removed after data cleaning due to insufficient information, leaving a total of 699 (Fig 2).

Table 1 presents the socio-demographic characteristics of the pre- and post-intervention study samples. The single largest group of women belonged to the 20–24 age group in both surveys, with slightly younger women represented in the post-intervention survey. The samples were similar for caste, religion and age at marriage for both surveys. The pre-intervention sample had a higher proportion of women who were illiterate (66.3%) compared to the post-intervention sample, where a higher proportion of women had achieved primary level education (54.5%). A higher proportion of women reported their husband's occupation as a farmer in the pre-intervention sample (60.5%), whereas in the post-intervention sample a higher

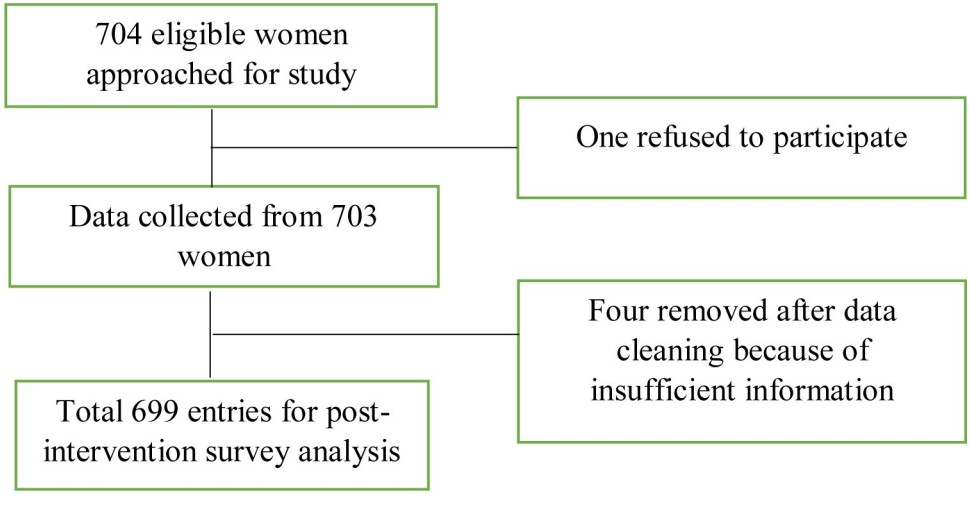

**Fig 2. Data analysis flow chart.**

**Table 1. Socio-demographic characteristics.**

| Characteristics | Pre-intervention N (%) | Post-intervention N (%) |
|---|---|---|
| **Age of women during study** | 420 | 699 |
| 15–19 | 46 (11.0) | 28 (4.0) |
| 20–24 | 163 (38.8) | 327 (46.8) |
| 25–29 | 148 (35.2) | 226 (32.3) |
| 30 and above | 63 (15.0) | 118 (16.9) |
| **Caste** | 420 | 699 |
| Disadvantaged | 380 (90.5) | 649 (92.8) |
| Advantaged | 40 (9.5) | 50 (7.2) |
| **Religion** | 420 | 699 |
| Hindu | 351 (83.6) | 587 (84.0) |
| Muslim and others | 69 (16.4) | 112 (16.0) |
| **Education** | 407 | 699 |
| Illiterate | 270 (66.3) | 205 (29.3) |
| Primary | 112 (27.5) | 381 (54.5) |
| Secondary and above | 25 (6.1) | 113 (16.2) |
| **Husband's occupation** | 420 | 699 |
| Farmer | 254 (60.5) | 234 (33.4) |
| Skilled labour and Teacher | 101 (24.0) | 141 (20.2) |
| Unskilled labour and Other | 65 (15.5) | 324 (46.4) |
| **Age at marriage** | 420 | 699 |
| Below 15 | 89 (21.2) | 124 (17.7) |
| 15–19 | 226 (53.8) | 374 (53.5) |
| 20 and above | 105 (25.0) | 201 (28.8) |
| **Total people living in house** | 420 | 685 |
| Less than 5 | 54 (12.9) | 172 (25.1) |
| 5–9 | 217 (51.7) | 313 (45.7) |
| 10–14 | 111 (26.4) | 159 (23.2) |
| 15–19 | 30 (7.1) | 32 (4.7) |
| 20 and above | 8 (1.9) | 9 (1.3) |
| **Total number of children** | 415 | 699 |
| Less than 3 | 364 (87.7) | 623 (89.1) |
| 3 and above | 51 (12.3) | 76 (10.9) |

proportion reported their husband's occupation as an unskilled labourer or others (46.4%). Looking at this comparative data from pre- and post-intervention survey, it is evident that these two sets of data are comparable but not the same. Some variables have improved from pre- to post-intervention survey such as literacy of women, with more women being educated to primary level education in post-intervention (54.5%) than pre-intervention survey (27.4%). While other variables changed slightly, the percentage of women aged 20–24 changed considerably from 38.8% to 46.8% (Table 1).

Most of the women in the pre-intervention sample gave birth at home (58.8%), but this proportion decreased in the post-intervention sample (29.3%) (Table 2). Similarly, women who gave birth at BCs increased significantly (from 2.4% to 28.3%). This was reflected in an increased proportion of births with skilled health professionals (increase from 53.7% to 70.5%). Women reported greater involvement in the decision about the birthplace, post-intervention (57.4%) and the number experiencing the optimal number of ANC visits (four and above) increased to 80.3% post-intervention.

**Table 2.  Health services, obstetric and maternal characteristics of respondents.**

| Characteristics | Pre-intervention (N, %) | Post-intervention (N, %) |
|---|---|---|
| **Birthplace** | 420 | 699 |
| Home | 247 (58.8) | 205 (29.3) |
| Birthing centre | 10 (2.4) | 198 (28.3) |
| Primary care facilities | 39 (9.3) | 88 (12.6) |
| Tertiary health centre | 124 (29.5) | 208 (29.8) |
| **Decision maker for birthplace** | 420 | 699 |
| Woman | 72 (17.1) | 102 (14.6) |
| Husband | 177 (42.1) | 86 (12.3) |
| Woman and family members | 13 (3.1) | 401 (57.4) |
| Family members/others | 158 (37.6) | 110 (15.7) |
| **Birth attendant** | 417 | 699 |
| Skilled health professionals | 224 (53.7) | 493 (70.5) |
| Unskilled people and others | 193 (46.3) | 206 (29.5) |
| **Received financial assistance for childbirth** | 413 | 693 |
| Yes | 105 (25.4) | 370 (53.4) |
| **Total number of pregnancies (gravida)** | 418 | 699 |
| 1–3 | 342 (81.8) | 586 (83.8) |
| 4 and above | 76 (18.2) | 113 (16.2) |
| **Frequency of antenatal check-up** | 373 | 699 |
| Less than 4 | 140 (37.5) | 138 (19.7) |
| 4 and above | 233 (62.5) | 561 (80.3) |

Several factors were significantly associated with the change between pre- and post-intervention surveys. These included: woman's age, woman's education, husband's education, iron tablets taken during pregnancy, tetanus toxoid (TT) vaccine received during pregnancy, money received for childbirth, birthplace, decision maker for birthplace, person assisting birth, number of ANC visits and knowledge if abortion is legal. Only these variables were entered in the multinomial regression analysis. The variables women's education, birthplace, decision maker for birthplace and satisfaction with childbirth services had a strong association with intervention (S1 Table).

Table 3 presents the adjusted multinomial regression analysis results for factors affecting choice of birthplace. Generally, controlling for all other variables, the likelihood of giving birth at a facility (either primary care facilities such as BCs or tertiary care facilities like hospitals) increased post-intervention (Table 3). The likelihood was only statistically significant for the primary care facilities (OR 5.60, p-value <0.001).

Women whose husbands or family members were the decision makers had an increased likelihood of having a facility birth. However, women were significantly less likely to give birth at either primary care facilities (OR 0.16, p-value <0.001) or hospitals/tertiary care facilities (OR 0.16, p-value <0.001) if they alone were responsible for deciding on the birthplace.

Respondents who reported a less than optimal number of ANC visits (one to three) compared to the recommended (four and over) had a significantly lower likelihood of giving birth at either primary care facilities (OR 0.39, p-value <0.001) or hospitals/tertiary care facilities (OR 0.63, p-value 0.021).

Generally, the likelihood of a facility birth decreased with the age of the respondent. The likelihood of health facility birth was significantly higher for age group 15–19 (OR 2.72, p-value 0.016 for primary care facilities and OR 3.02, p-value 0.004 for hospitals/tertiary care facilities), which declined but remained significant for age group 20–24.

**Table 3. Adjusted multinomial logistic regression of factors affecting place of delivery.**

| Variables | Primary care facility vs home | | Hospitals/tertiary vs home | |
|---|---|---|---|---|
| | OR (95% CI) | p value | OR (95% CI) | p value |
| Intervention (Ref Pre) | | | | |
| Post | 5.60(3.34,9.38) | <0.001 | 1.56 (0.98,2.47) | 0.060 |
| Decision maker for birthplace (Ref Family members/others) | | | | |
| Women | 0.16(0.08,0.29) | <0.001 | 0.16(0.08,0.30) | <0.001 |
| Husband | 3.17 (1.87,5.37) | <0.001 | 2.80(1.75,4.47) | <0.001 |
| Women & family members | 1.76 (1.08,2.85) | 0.023 | 1.78 (1.10,2.88) | 0.020 |
| Frequency of ANC visit (Ref 4 and above) | | | | |
| Less than 4 (1–3) | 0.39 (0.26,0.60) | <0.001 | 0.63 (0.43,0.91) | 0.014 |
| Age (years) (Ref 30 and above) | | | | |
| 15–19 | 2.72 (1.20,6.17) | 0.016 | 3.02(1.42,6.44) | 0.004 |
| 20–24 | 1.64 (1.01,2.68) | 0.045 | 2.28(1.40,3.70) | 0.001 |
| 25–29 | 1.27 (0.76,2.10) | 0.355 | 1.25 (0.75,2.10) | 0.394 |
| Education (Ref Secondary and above) | | | | |
| Illiterate | 0.79 (0.43, 1.43) | 0.438 | 0.66 (0.36,1.20) | 0.169 |
| Primary | 0.49 (0.28,0.87) | 0.014 | 0.58 (0.33,1.03) | 0.063 |
| Husband's occupation (Ref Skilled labourer & teacher) | | | | |
| Farmer | 0.86 (0.57,1.27) | 0.447 | 0.77 (0.52,1.15) | 0.196 |
| Unskilled labourer/others | 0.86 (0.54,1.38) | 0.530 | 1.11 (0.70,1.74) | 0.665 |
| Knowledge if abortion is legal (Ref No) | | | | |
| Don't know | 0.61(0.37,0.99) | 0.046 | 0.90 (0.55, 1.49) | 0.683 |
| Yes | 0.72 (0.42, 1.22) | 0.217 | 0.96 (0.56, 1.64) | 0.867 |
| Money received for childbirth (Ref No) | | | | |
| Don't know | 0.92 (0.07, 11.039) | 0.947 | 1.66 (0.16, 17.14) | 0.672 |
| Yes | 0.46 (0.32, 0.66) | <0.001 | 0.48 (0.34, 0.68) | <0.001 |

Women's education also affected the birthplace. Women who had only attended primary level education were half as likely to give birth at primary care facilities (OR 0.49, p-value 0.014), compared to those with 'secondary level education and above'. Although there was a difference in relation to tertiary level care (hospital), this was not statistically significant.

Women were less likely to give birth at health facilities even when they received money for childbirth compared to those who did not receive money. The results were significant for both primary care facilities (OR 0.46, p-value <0.001) and hospitals/tertiary care facilities (OR 0.48, p-value <0.001).

## Discussion

Health promotion interventions designed to increase access and utilisation of perinatal care facilities have been recommended by Smith et al. [20]. This paper reports an intervention that increased the births at BCs and decreased home births. The intervention also had an influence on women's autonomy and the use of perinatal care facilities at BCs. The results indicate that if women were included in the decision making about place of birth, they were more likely to give birth at health facilities. Women's level of education had an influence on determining where they would give birth and if they would use perinatal care facilities available at health facilities. Having four ANC visits was also reported as important factor in choosing health facilities for childbirth.

## Increased birth at birthing centres

This study demonstrates that an intervention promoting BCs has the potential to increase the proportion of women birthing in a health facility and decrease the proportion of home births. The increase in BCs births exceeded the national average of 27% 2016, the latter decreased from 29% in 2015[10].

Skilled care during pregnancy and childbirth can be achieved by safe and clean delivery at birth and care of the newborn at birth [28]. Giving birth at health facilities not only prevents/treats pregnancy related complications but also helps in reducing maternal and neonatal mortality [29, 30]. Thus, in low-income countries such as Nepal, it is preferable to reduce home births in the absence of a SBA and increase institutional births where a SBA will be in attendance. Encouraging BC birth is the best way to secure improved SBA attendance in rural communities. This is also the policy of the Government of Nepal which launched free institutional delivery care in 2009 [31]. In addition, the government's policy of upgrading BCs and strengthening the competency of health staff may be helpful in increasing institutional delivery rates [32]. This study also shows that an intervention of supporting BCs has effectively decreased the number of home births without a SBA and increased the number of births at these BCs.

## Health promotion intervention

The change in birthplace from home to health facilities in this study can be explained as the effect of health promotion programme conducted by local health promotors targeting local women as part of the intervention. The intervention took into account the diverse/ changing needs of local communities and the best use of existing resources [33]. Review studies have shown community-based intervention packages reduce morbidity for women, mortality and morbidity for babies and improves care-related outcomes and the health of mothers, neonates and children, particularly in low and middle-income countries [34, 35]. One study also highlighted the value of integrating maternal and newborn care in community settings through a range of interventions which could be effectively delivered through community health workers and health promoters [34]. A review concluded community-based interventions could be an important component of a comprehensive approach to accelerating improvements in maternal health and reducing preventable maternal deaths by 2030 [36].

It is important that health promotional interventions are targeted at women, their husbands and family members, since the results of this study show that a majority of the decisions related to childbirth and maternity care is taken by husbands and family members especially the mother-in-law. In the Nepalese context, women have less control over decisions related to birth processes; for example, in going for ANC visits [37]. Therefore, it is important that midwives work in partnership with mothers and families, especially mothers-in-law, thus facilitating decisions about the care they need [38].

The results also highlighted drastic increase in financial assistance received for childbirth through the 'Aama programme' (a kind of financial incentive scheme specifically for women who gives birth at health facilities) [13] pre and post intervention. The rapid increase in percentages here might be attributed to health promoters' role in promoting the 'Aama Programme' during the meetings conducted with women's group and mothers' groups. However, more research is required on this as the results of regression analysis demonstrated less likelihood of giving birth in health facilities when women received financial assistance compared to when women did not receive any assistance.

## Women's autonomy

Women autonomy was seen as an important factor that determined the uptake of health facilities (the BCs). Determinants of women's autonomy, such as making the decision around birthplace, were important factors affecting choice of birthplace. The results of multinomial regression analysis showed that when women solely decided about their birthplace, they were less likely to attend a health facility for childbirth. The reason behind this could be that these women many not have anyone else to depend upon such as their mother-in-law or if their husband is a migrant worker. Another reason could be that they belong to lower socio-economic strata and did not feel they had the resource for a facility birth. The "Aama programme" provides certain amount but would not cover all of the costs. Conversely, when women were included in decision, but were not the sole decision-makers about where to give birth, they were more likely to have a facility birth. Research has shown that although women want to choose their birthplace based on safety and other grounds [39], for many women the decision to give birth at a health facility is not their own but involves their family as well as the community [40]. The women sometimes find that their right to choose their birthplace is compromised because of cultural and traditional practices [41]. A study in rural Nepal also established that the decision for uptake of the institutional birth services was influenced more by family members or family members and women and not by women alone [42]. Similarly, husbands' control over decision making regarding the birthplace was found in Tanzania [43] and Bangladesh [44].

The results of this study identified the need for involving women in the decision-making process including choosing their place of childbirth. Involving women in the decisions on maternal healthcare, including choosing the birthplace, ensures that women are empowered and can exercise their rights over reproductive healthcare. The findings suggest there is a need for further work focusing on educating mothers about the importance of giving birth at health facilities along with educating husbands and other family members. This should include the importance of involving women in decision making regarding their healthcare and specifically about where to give birth.

## Women's literacy level

The education level of women determined if they utilised the birth services at BCs. A study in Ethiopia found that women's educational level affected the birthplace, but not that of their husbands [45]; however, this contrasted to the study in rural Nepal where the educational status of women had no effect on deciding the birthplace [42].

## Importance of having optimum ANC visits

The results of multinomial regression analysis showed a decreased likelihood of giving birth at primary care facilities and hospitals/tertiary care facilities if the women had less than four ANC visits with reference to 'four and above ANC visits'. The results thus depict the importance of having four or more ANC visits, which indicates that women are generally more concerned about their babies' wellbeing in addition to encouraging women to attend health facilities for giving birth. Studies in Nepal have highlighted the importance of education, socio-economic and socio-cultural status on the uptake of ANC. This pointed to the presence of cultural barriers for Terai women to attending the ANC visits [46]. Additionally, decision making power related to ANC visits was less for women in Terai region compared to those living in mountains and hilly regions [47]. The literature suggests that people living in the mountains and inner Terai (Nawalparasi lies in inner Terai) regions are an absolute minority and belong to most marginalised groups [47]. Furthermore, one study found that improving the

quality of ANC visits will have a positive and motivating effect on women utilising institutional delivery services [48]. A study in Nepal has shown that FCHVs play a pivotal role in improving antenatal care [49] and this will also be the case in this study specifically due to involvement of health promoters who worked together with FCHVs. Similar to the above-mentioned studies, the population of this study consisted mainly of women belonging to disadvantaged castes in the Terai, with low levels of education and decision-making power. These women are dependent on either their husbands or other family members for decisions related to household and health related matters.

## Conclusion

BCs have the potential to increase the proportion of women who have access to a skilled birth attendant. The uptake of BC care is a complex issue, but this study has shown that the role of health promoters is important in rural Nepal.

## Supporting information

**S1 Table. Association of intervention with other variables.**
(DOCX)

## Acknowledgments

We would like to thank all the participants who took part in this study. Similarly, our sincere thanks to Green Tara Nepal for providing technical assistance for the fieldwork in this study.

## Author Contributions

**Conceptualization:** Preeti Mahato, Edwin van Teijlingen, Catherine Angell, Vanora Hundley.

**Data curation:** Preeti Mahato, Edwin van Teijlingen.

**Formal analysis:** Preeti Mahato, Edwin van Teijlingen, Padam Simkhada, Catherine Angell.

**Funding acquisition:** Preeti Mahato, Edwin van Teijlingen, Padam Simkhada.

**Investigation:** Preeti Mahato, Edwin van Teijlingen, Padam Simkhada, Catherine Angell.

**Methodology:** Preeti Mahato, Edwin van Teijlingen, Padam Simkhada, Catherine Angell, Vanora Hundley.

**Project administration:** Preeti Mahato, Vanora Hundley.

**Resources:** Preeti Mahato, Vanora Hundley.

**Software:** Preeti Mahato.

**Supervision:** Edwin van Teijlingen, Padam Simkhada, Catherine Angell.

**Validation:** Edwin van Teijlingen, Vanora Hundley.

**Visualization:** Edwin van Teijlingen, Vanora Hundley.

**Writing – original draft:** Preeti Mahato.

**Writing – review & editing:** Preeti Mahato, Edwin van Teijlingen, Padam Simkhada, Catherine Angell, Vanora Hundley.

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
