## [Decision Letter · Decision Letter 0]

13 Mar 2020

PONE-D-19-33046

A study of birthing centres and maternity users in southern Nepal

PLOS ONE

Dear Dr Mahato,

Thank you for submitting your manuscript to PLOS ONE. After careful consideration, we feel that it has merit but does not fully meet PLOS ONE’s publication criteria as it currently stands. Therefore, we invite you to submit a revised version of the manuscript that addresses the points raised during the review process.

The reviewers have provided some really helpful points to improve the overall quality of the manuscript. Among these is the request for more detail on the intervention and clarification of choice of reference value in the multinomial analyse. 

We would appreciate receiving your revised manuscript by Apr 27 2020 11:59PM. To enhance the reproducibility of your results, we recommend that if applicable you deposit your laboratory protocols in protocols.io, where a protocol can be assigned its own identifier (DOI) such that it can be cited independently in the future. For instructions see: http://journals.plos.org/plosone/s/submission-guidelines#loc-laboratory-protocols

We look forward to receiving your revised manuscript.

Kind regards,

Christine E East

Academic Editor

PLOS ONE

Journal Requirements:

2) Please provide additional details regarding participant consent. In the ethics statement in the Methods and online submission information, please ensure that you have specified (1) whether consent was informed and (2) what type you obtained (for instance, written or verbal, and if verbal, how it was documented and witnessed). If your study included minors, state whether you obtained consent from parents or guardians. If the need for consent was waived by the ethics committee, please include this information.

3) We note that you have stated that you will provide repository information for your data at acceptance. Should your manuscript be accepted for publication, we will hold it until you provide the relevant accession numbers or DOIs necessary to access your data. If you wish to make changes to your Data Availability statement, please describe these changes in your cover letter and we will update your Data Availability statement to reflect the information you provide.

Reviewers' comments:

Reviewer's Responses to Questions

**Comments to the Author**

1. Is the manuscript technically sound, and do the data support the conclusions?

Reviewer #1: Yes

Reviewer #2: Yes

Reviewer #3: Partly

2. Has the statistical analysis been performed appropriately and rigorously? 

Reviewer #1: Yes

Reviewer #2: Yes

Reviewer #3: No

3. Have the authors made all data underlying the findings in their manuscript fully available?

Reviewer #1: Yes

Reviewer #2: Yes

Reviewer #3: Yes

4. Is the manuscript presented in an intelligible fashion and written in standard English?

Reviewer #1: Yes

Reviewer #2: Yes

Reviewer #3: No

5. Review Comments to the Author

Reviewer #1: General Comments:

This article addresses a very pertinent issue in the field of maternal and child health in Nepal—birthing centers. Birthing centers have been developed and heavily promoted over the last decade by the Ministry of Health and Population of Nepal. Some function very well and have high delivery volumes but many are underutilized. The authors show that an intervention aimed at improving the birthing center and promoting it in the community through FCHVs and health promoters is associated with an increase in uptake in facility births.

The overall study is sound with a pre and post intervention component and multinomial regression analysis of predictors of place of delivery. However, one topic which I would like the authors to explore further is the issue of the “Aama program” which gave financial incentives to mothers for health facility delivery. The pre and post intervention Table 2 shows a large increase in percentage of mothers receiving financial assistance for childbirth. (I am presuming this is from the government’s Aama program). However, in the multinomial regression, this variable is not appropriately explored—the “money received for childbirth reference category is “Don’t know”. I believe either “yes” or “no” should have been used.

Specific Comments

Line 75: Should be “healthy women”

Line 109-111: “This paper reports a study which…” Very vague—please lay out the specific objectives of your study here.

Line 116-123: Although you provide some level of detail regarding the intervention, I wanted more detail. For example, how much financial support was given to each birthing center? Was this strictly through the NGO or through the government? Regarding the health promotion program—is it not part of the FCHVs role to educate mothers about ANC/PNC, sanitation, etc. What exactly did your program do? How many mother-in-law meetings were conducted?

Line 185-187: “While other variables changed slightly such as age of marriage for women remained higher for age group 20-24 in both pre- and post-intervention survey although the percentage changed from 38.8% to 46.8%.

-This is not a complete sentence.

-I also don’t see the numbers 38.8% and 46.8% in the Tables.

Line 197 (Table 2): I think it’s quite interesting that the “Received financial assistance for childbirth” was so different pre and post intervention (25.4% to 53.4%). I wonder how much of the increase in BC births was due to this financial assistance program (“Aama program”) versus your intervention.

Table 3 comments:

-Please fix formatting for “Decision maker for birthplace” section: the numbers should be moved up a row”

-For the “knowledge if abortion is legal” variable, why did you use “Don’t know” as the reference category? Why not either “Yes or No”?

-For the “Money received for childbirth” variable, why did you use “Don’t know” as the reference. I don’t think this makes sense. Your pre and post intervention Table 2 clearly shows a huge jump in this variable. I think that if you used either the “yes” or “no” category as the reference, you would have very significant results. It might change your whole Table 3.

-I am curious about why you used the “Time baby first washed” variable. Why is this important?

Line 282: as part of “the” intervention

Line 289: “A” review study concluded..

Line 293: show that “a” majority of

Line 294: “the” mother-in-law

Women’s autonomy section

Lines 299-313:

I’m not sure that I understand your argument about women’s autonomy. I feel that your numerical results are the opposite of your argument. You state, “The results of multinomial regression analysis showed that when women solely decided about their birthplace, they were less likely to attend a health facility for childbirth.” In a later paragraph you state, “Involving women in the decisions related to their maternal healthcare including choosing the birthplace ensures that women are empowered and can exercise their rights over reproductive healthcare.” These seem polar opposite arguments to me. It seems that if only the woman chooses, she chooses not to deliver at a health facility. Am I misunderstanding something?

I wonder if one way to explain the discrepancy in the results is to surmise that when only the woman is solely involved in the decision, she may not have anyone else to depend upon (perhaps, there is no mother-in-law, perhaps husband is a migrant worker, etc). I wonder if these women were also of lower socioeconomic strata and did not feel they had the resources for a facility birth—the Aama program gives them a certain amount but would not cover all of the costs.

Reviewer #2: This study evaluated an intervention to increase access and utilisation of perinatal care facilities in community settings. The study is well written, however, I have a few concerns:

Minor revisions

1) In line 58, sentence with ‘Measuring SBA rates’ is not clear.

2) Abbreviations: “BC” in line 72 should instead be in line 70 where birthing centre is written for the first time. In line 121, ANC and PNC have been written first time but with abbreviations.

3) Line 77 mentions that basic obstetric care are available from ‘Sub health posts (SHPs)’. SHPs have been upgraded to Health Posts (HPs). Please remove SHP from subsequent content as well.

4) Hospitals are no longer categorized as regional and zonal. Hence, in line 82 revise the category of current hospitals correctly.

5) Lines 88-93. ‘ANMs providing care especially at BCs’ is confusing because ANMs work also in other Health Posts that are not birthing centres, as well as in hospitals which you have mentioned in last sentence. So, ‘In most of the BCs ANMs are service providers’ would be more appropriate.

6) Line 119. Who were ‘health promoters’?

7) Figure 1. How many wards from each village development committee were selected and what was the basis for? Explanation of the figure is required.

8) Line 149. Which data were analysed as ‘secondary data’?

9) In Table 2, variable ‘Birth Attendant’ would be better than ‘Skilled Birth Attendant’.

10) Your analysis and results would be more clear if ‘outcome variables’ of the intervention are clearly mentioned (the characteristics in table 2 seem to be the ones) and assessed for statistical significant difference in the outcome variables between pre and post intervention, as well as their strength of association with intervention. And then another table with contributing factors to place of birth would be sufficient.

11) The manuscript should be read thoroughly and corrected for few errors in English writing before submitting revised version.

Reviewer #3: Title

Revise the title to reflect the aim of the study—the evaluation of the intervention as stated in the manuscript. I would say ‘Impact of …………..interventions in improving maternity services in rural Nepal’.

Abstract

BCs are supposed to provide care for women without complications. However, this is not the case, more importantly in the rural areas. Women often present late, and therefore mostly with a complication. An Skilled Birth Attendant is supposed to manage most of the Basic Emergency Obstetric Complications.

Introduction

As this is an evaluation study of BC, I would start the introduction section with paragraph third. The introduction as of now is a bit general and does not clearly demonstrate a review of prior interventions to support birthing centres, gaps in those interventions and the need for doing this particular study.

Materials and Methods

I would again emphasize to clearly describe the interventions carried out in the two birthing centres and 4 catchment VDCs focussing on:

o When did the two health facilities (the BCs as mentioned now) started functioning (or perhaps named/announced) as birthing centres?

o What specific interventions did we do in addition to existing care/support in the selected BCs?

o What kind of prior health promotion interventions existed before?

o What did we do in the 4 communities?

o Did we hire additional staffs, SBAs? Community health workers?

o Any new trainings? Supplies?

o When did the intervention start? How did we monitor the intervention adherence/fidelity?

o What were primary outcomes of interest?

o What were other/secondary outcomes of interest?

Please briefly describe the qualification of the enumerators including their gender, number

Figure 1. mention the total wards, number of wards selected randomly

Did you know the list of households selected in each ward? It would be great to mention what is the proportion of the approached household (704) of the total households in the ward?

Please mention briefly about what you found after cross checking 10% sample.

You approached 704 for the post-intervention survey. How many did you approach in pre-intervention survey? Table 1 shows 420 included in Analysis? How such a huge difference in size of the participants for pre and post intervention survey?

Who approached these women? How did you recruit them?

Authors nowhere talk about consents form participants and ethical approval for this study. This is very important and I want to clearly know whether this study was ethically approved. Please mention it.

Results

Table 2. where do you demonstrate the impact of the birthing centre support intervention as aimed in this study? How do you justify that the intervention worked?

Why did you also include hospital when the focus was to see the impact of BC intervention?

Discussion

You stated ‘’ This paper reports an intervention that increased the births at BCs and decreased home births. The intervention also had an influence on women’s autonomy and the use of perinatal care facilities at BCs.” I would again like to know the intervention—what exactly was done in the 2 BCs and 4 villages?.

How do you discuss the contribution of already exiting network of female community health volunteers, mothers groups, the voucher programme (monetary incentives to motivate women to come for antenatal and childbirth)?

Others

Authors mention two NGOs providing technical support, and also describe an NGO involved in delivering the intervention. Please clearly mention as disclaimer whether this was a funded study?

6. PLOS authors have the option to publish the peer review history of their article (what does this mean?). If published, this will include your full peer review and any attached files.

Reviewer #1: No

Reviewer #2: Yes: Rajani Shah Malla

Reviewer #3: No

---

## [Author Response · Author response to Decision Letter 0]

15 Apr 2020

Reviewer 1

Article addresses pertinent issue in the field of maternal and child health in Nepal—birthing centers. … Authors show that an intervention aimed at improving the birthing center and promoting it in the community through FCHVs and health promoters is associated with an increase in uptake in facility births. Thank you 

The overall study is sound with a pre and post intervention component and multinomial regression analysis of predictors of place of delivery. … I would like authors to explore further is the issue of the “Aama program” which gave financial incentives to mothers for health facility delivery. The pre and post intervention Table 2 shows a large increase in % mothers receiving financial assistance for birth. (I presume this is from government’s Aama program). However, in multinomial regression, this variable is not appropriately explored—the “money received for childbirth reference category is “Don’t know”. I believe either “yes” or “no” should have been used. We have provided details on “Aama Program” in Discussion. See lines 296-301.

We understand that the reference category could have been ‘yes’ or ‘no’ but since the reference category was always chosen the last category for convenience reason, we did not anticipate this could have made a lot of difference in the results. Rather we viewed that changing reference category might only change how results were interpreted. For example, if ‘no’ was selected as reference category then it would be interpreted compared to those who received financial assistance (Table 2). 

Line 75: Should be “healthy women” This is changed now

Line 109-111: “This paper reports a study which…” Very vague—please lay out the specific objectives of your study here. Line 110-113 now changed to “This paper evaluated the effects of an intervention consisting of supporting BCs and community-based health promotion programme on increasing access and utilisation of perinatal care facilities in community settings.” 

Line 116-123: Although you provide some detail regarding the intervention, I wanted more, e.g. how much financial support was given to each BC? Was this strictly through the NGO or through the government? Regarding the health promotion program—is it not part of the FCHVs role to educate mothers about ANC/PNC, sanitation, etc. What exactly did your program do? How many mother-in-law meetings …d? We have provided additional details about the intervention and mentioned that the intervention and pre-intervention survey was carried out by a local NGO. We conducted post-intervention survey and received the pre-intervention survey data from the NGO (lines 119-137). 

Line 185-187: “While other variables changed slightly such as age of marriage for women remained higher for age group 20-24 in both pre- and post-intervention survey although the percentage changed from 38.8% to 46.8%.

-This is not a complete sentence. Line 206-208.

This sentence has been changed to “While other variables changed slightly, the percentage of women aged 20-24 changed considerably from 38.8% to 46.8%. 

I also don’t see the numbers 38.8% and 46.8% in the Tables These are in Table 1, age group of women during pre-intervention (38.8%) and post-intervention (46.8%). This has been clarified in the text (line 208).

Line 197 (Table 2): I think it’s quite interesting that the “Received financial assistance for childbirth” was so different pre and post intervention (25.4% to 53.4%). I wonder how much of increase in BC births was due to this financial assistance (“Aama program”) versus your intervention Financial incentive was from “Aama program” and not from our intervention. These figures show the percentage change that reported the receiving of financial incentive pre and post intervention. Added few sentences about this in the Discussion section (see lines 296- 301).

Table 3 fix formatting for “Decision maker for birthplace” section: numbers should be moved up a row” Thank you we have done this in clean copy

For the “knowledge if abortion is legal” variable, why did you use “Don’t know” as the reference category? Why not either “Yes or No”? The ‘don’t know’ was used as reference category for convenience reason as it was the last category. Also, the number who didn’t know that abortion was legal was quite high.

For the “Money received for childbirth” variable, why did you use “Don’t know” as the reference. I don’t think this makes sense. Your pre and post intervention Table 2 clearly shows a huge jump in this variable. I think that if you used either the “yes” or “no” category as the reference, you would have very significant results. It might change your whole Table 3. Again ‘don’t know’ was used as reference category for convenience reason since this was the last category. There were no natural reference categories and we did not anticipate that changing reference categories might make lot of difference to the results. Rather we viewed that changing reference category might only change how results are interpreted.

I am curious about why you used the “Time baby first washed” variable. Why is this important? We understand removing this variable from the table will reduce confusion, therefore we have done this.

Line 282: as part of “the” intervention

Line 289: “A” review study concluded.

Line 293: show that “a” majority of

Line 294: “the” mother-in-law Thank you we have changed these now.

Women’s autonomy section Lines 299-313:

I’m not sure I understand your argument about women’s autonomy. I feel that your numerical results are the opposite of your argument. You state, “The results of multinomial regression analysis showed that when women solely decided about their birthplace, they were less likely to attend a health facility for childbirth.” In a later paragraph you state, “Involving women in the decisions related to ……..their rights over reproductive healthcare.” These seem polar opposite arguments….. if only the woman chooses, she chooses not to deliver at a health facility. Am I misunderstanding?

One way to explain the discrepancy in results is to surmise that when only woman is solely involved in decision, she may not have anyone else to depend upon (perhaps, there is no mother-in-law, perhaps husband is a migrant worker, etc). I wonder if women were also of lower socioeconomic strata and felt they had no resources for a facility birth—the Aama program gives them a certain amount but would not cover all of the costs. Thank you, as you noted, when women solely decided about their birthplace, they were less likely to attend a health facility for childbirth, could mean that when they decide by themselves they choose not to deliver at health facility. 

We also agree with your comments that these women might not have any support or maybe of low socioeconomic status. In addition, we also mention that such women need further education and awareness about importance of giving birth at a health facility along with husbands and other family members. This has been added to the discussion (line 307-311).

Reviewer 2

In line 58, sentence with ‘Measuring SBA rates’ is not clear. Changed this to “Measuring and monitoring of SBA remains a challenge because of the wide variety of definitions used.” (line 59).

Abbreviations: “BC” in line 72 should instead be in line 70 where birthing centre is written for the first time. In line 121, ANC and PNC have been written first time but with abbreviations. Thank you, corrected this.

Line 77 mentions that basic obstetric care … from ‘Sub health posts (SHPs)’. SHPs have been upgraded to Health Posts (HPs). Please remove SHP from subsequent content as well. Thank you, we have done this.

Hospitals are no longer categorized as regional and zonal. Hence, in line 82 revise the category of current hospitals correctly. We have mentioned as “central, provincial and district” (line 83).

Lines 88-93. ‘ANMs providing care especially at BCs’ is confusing as ANMs work also in other Health Posts that are not BCs, as well as in hospitals as you have mentioned in last sentence. So, ‘In most of the BCs ANMs are service providers’ would be more appropriate. We have added this sentence “In most of the BCs, Auxiliary Nurse Midwives (ANMs) provide much of the primary care maternity services in Nepal.” (line 89).

Line 119. Who were ‘health promoters’? The health promoters were employed by the NGO and trained to provide health promotion program in their local community. Now mentioned in the text as “The health promotion programme consisted of training local health promoters employed by the NGO, who then trained Female Health Community Volunteers (FCHVs).” (see lines 126-128).

Figure 1. How many wards from each village development committee were selected and what was the basis for? Explanation of the figure is required. We added: “In order to get a spread on poorer and slightly better off wards as well as those closer to the BCs and those further away 29 out of 36 wards were visited.” (lines 150-52).

Line 149. Which data were analysed as ‘secondary data’? The secondary data were the pre-intervention survey data made available by the NGO. The pre-intervention survey was conducted by the NGO as is mentioned in line 158-160.

In Table 2, variable ‘Birth Attendant’ would be better than ‘Skilled Birth Attendant’. Thank you, we have changed this.

Your analysis and results would be more clear if ‘outcome variables’ of the intervention are clearly mentioned (the characteristics in table 2 seem to be the ones) and assessed for statistical significant difference in the outcome variables between pre and post intervention, as well as their strength of association with intervention. And then another table with contributing factors to place of birth would be sufficient. The outcome variables of the intervention are birth at BCs (primary outcome), number of ANC, women’s decision making about place of birth and satisfaction with childbirth services (secondary outcome) (see lines 169-172). Additional analysis such as association of intervention with other variables was undertaken and their strength of association is shown in S1 Table. It is supplementary, if editor/reviewers prefer it can go in main text.

The manuscript should be corrected for few errors in English writing before resubmitting. The manuscript has been proof-read and errors corrected

Reviewer 2

Revise the title to reflect the aim of the study—the evaluation of the intervention as stated in the manuscript. I would say ‘Impact of …………..interventions in improving maternity services in rural Nepal’. Thank you for suggestion. We evaluated an intervention but did not carry it out, therefore we changed the title to: “Evaluation of a health promotion intervention associated with birthing centres in rural Nepal”

Abstract

BCs are supposed to provide care for women without complications. However, this is not the case, more importantly in the rural areas. Women often present late, and therefore mostly with a complication. An Skilled Birth Attendant is supposed to manage most of the Basic Emergency Obstetric Complications. Thank you, in our experience women often present late but that does not mean they have complications. Those women who have complications are referred to higher level health facilities. Since most women even in rural areas of Nepal go for at least one ANC, the ANMs are aware of complications and if there is complication, the ANMs always refer these women to hospital as they do not want to take risk.

Introduction

As this is an evaluation study of BC, I would start the introduction section with paragraph third. The introduction as of now is a bit general and does not clearly demonstrate a review of prior interventions to support birthing centres, gaps in those interventions and the need for doing this particular study. Thank you for your comments. We understand that this is evaluation study, but we also need to set the scene for a general audience. So, we feel it is important to start with a general introduction about SBA, moving to maternity care provision in Nepal. We do have a paragraph related to interventions but since this is not an intervention study but evaluation study, we thought this can go in the third paragraph.

Materials and Methods

I would emphasize to clearly describe the interventions carried out in the two BCs and 4 catchment VDCs focussing on:

When did the two health facilities (BCs mentioned now) started functioning (or perhaps named/announced) as BCs? We have mentioned 2015 and 2016. (line 122)

What specific interventions did we do in addition to existing care/support in the selected BCs? Health promotion and supporting BCs (added to line 122-137)

What kind of prior health promotion interventions existed before? None except the health promotion role of FCHVs (sentence added to line 130-131.)

What did we do in the 4 communities? We have mentioned this in Materials and Methods: “the health promoters conducted meetings with mothers-in-laws as a strategy for creating demand for utilisation of the BCs [24]. For example, in 2016, there were 157 mothers-in-law meetings and 334 women’s group meetings. The health promoters and FCHVs met mother groups on a monthly basis and discussed various issues related to women’s health through a curriculum covering content on ANC/PNC, baby feeding, sanitation and hygiene.” (lines 131-137)

Did we hire additional staffs, SBAs? Community health workers? Yes, the NGO hired additional staff also called as health promoters who were trained by the NGO for their role. Added to line 127.

Any new trainings? Supplies? We have mentioned “The support included refurbishing the health facilities infrastructure, providing equipment required for normal delivery, and training all the ANMs at these two BCs and appointing two local health promoters” (lines 122-128)

When did the intervention start? How did we monitor the intervention adherence/fidelity? We added: “The intervention, that was conducted by a local non-governmental organisation (NGO) during 2014-16, involved supporting two BCs and conducting a health promotion programme with local women.” (lines 119-120).

What were primary outcomes of interest?

What were other/secondary outcomes of interest? Primary outcome originally was birth at BCs and secondary outcome included number of ANC, women’s decision making about place of birth and satisfaction with childbirth services. Primary outcome changed to place of birth consisting of various categories i.e. home, primary care facilities including BCs and tertiary care facilities. This became clear when we tried to conduct chi square and regression analysis as descriptive findings of survey analysis showed that data related to place of birth consisted of three categories as mentioned above (see line 169-174).

Please briefly describe the qualification of the enumerators including their gender, number Post-intervention data were collected by eight female data enumerators having at least a degree level qualification in a health subject who were trained for two days by the first author. This has been added to lines 162-164.

Figure 1. mention the total wards, number of wards selected randomly

 Now added: “In order to get a spread on poorer and slightly better off wards as well as those closer to the BCs and those further away 29 out of 36 wards were visited.” (lines 150-152.

Did you know the list of households selected in each ward? It would be great to mention what is the proportion of the approached household (704) of the total households in the ward? Since this was a household survey, we approached all eligible participants from each household of selected wards of 4 VDCs and whoever agreed to take part was recruited. Very few refused to take part and response rate is more than 99% (One refused to take part in post-intervention survey) (Figure 2).

Please mention briefly about what you found after cross checking 10% sample. The main aim of cross checking 10% sample was to check completeness of data entry. The cross checking identified a small number of discrepancies in data entry and these were corrected by the first author who then used this to supervise the trainee for future data entry. This has been added to lines 165-167.

You approached 704 for the post-intervention survey. How many did you approach in pre-intervention survey? Table 1 shows 420 included in Analysis? How such a huge difference in size of the participants for pre and post intervention survey? As mentioned, pre-intervention survey was conducted by the NGO. For post-intervention survey, we approached 704, out of which one refused to take part and four removed after data cleaning leaving 699 entries. More participants were approached for post-intervention survey for more detailed sub-group analyses to ensure that sub-populations could be compared and linked to qualitative part of study not reported here (see Figure 2).

Who approached these women? How did you recruit them? Data enumerators approached women and if they agreed a structured questionnaire was completed by them (lines 149-150).

Authors nowhere talk about consents form participants and ethical approval for this study. …. Please mention it. Thank you for noting this, we have ethical approval. Now mentioned in materials and methods section (lines 180-186).

Results

Table 2. where do you demonstrate the impact of the birthing centre support intervention as aimed in this study? How do you justify that the intervention worked? Conducting adjusted regression allowed adjustment of all other factors so that the effects of certain variable are measured. This also ensures that the results obtained is not just a temporal trend but is likely the effect of intervention.

Why did you also include hospital when the focus was to see the impact of BC intervention? Primary outcome was originally birth at BCs but changed to place of birth consisting of three categories: home, primary care (incl. BCs) and tertiary care facilities. This became clear when we tried to conduct chi square and regression analysis as the descriptive findings of survey analysis showed that data related to place of birth consisted of three categories as mentioned above (see lines 169-174).

Discussion

You stated ‘’ This paper reports an intervention that increased the births at BCs and decreased home births. The intervention also had an influence on women’s autonomy and the use of perinatal care facilities at BCs.” I would again like to know the intervention—what exactly was done in the 2 BCs and 4 villages? We have provided description on intervention as “The intervention, that was conducted by a local non-governmental organisation (NGO) during 2014-16, involved supporting two BCs and conducting a health promotion programme with local women in four VDCs. These two BCs started functioning in the year 2015 and 2016. The support to BCs was provided in the form of refurbishing health facilities infrastructure, providing equipment required for normal delivery, recruiting two local ANMs for two years, training all the ANMs at these two BCs and appointing health promoters who were trained for their role. ……….. The health promoters and FCHVs also met mother groups on a monthly basis and discussed various issues related to women’s health through a curriculum covering content on ANC/PNC, baby feeding, sanitation and hygiene. The classes were informal and participatory, lasting about 1-2 hours [24].” (lines 119-137).

How do you discuss contribution of already existing network of FCHVs, mothers groups, voucher programme (monetary incentives to motivate women to come for ANC & birth)? We have mentioned about the contribution of FCHVs and Aama Programme in Discussion section (see lines 296-301, 307-311). 

Others

Authors mention two NGOs providing technical support, and also describe an NGO involved in delivering the intervention. Please clearly mention as disclaimer whether this was a funded study? A local NGO conducted the intervention and provided technical support during post-intervention survey. This is a part of a PhD study at UK university with technical support from NGO but no financial support from the NGO. See also the Acknowledgments.

---

## [Editor Report · Decision Letter 1]

28 Apr 2020

PONE-D-19-33046R1

Evaluation of a health promotion intervention associated with birthing centres in rural Nepal

PLOS ONE

Dear Dr Mahato,

Thank you for submitting your revised manuscript to PLOS ONE. Most of the revisions are acceptable. After careful consideration, we feel that it has merit but does not fully meet PLOS ONE’s publication criteria as it currently stands. Therefore, we invite you to submit a revised version of the manuscript that addresses the points raised during the review process.

Abstract

- Please add numeric results to the results section

-Avoid stating “significantly more likely”. The post-intervention survey provided evidence that women were more likely to … than prior to the intervention.

In the tracked copy, P5. Line 73. Need to start a sentence with a word, not an abbreviation.

Table 3. This needs the number and percent for each response. Some of these are already in Table 1. However, important factors such as knowledge about abortion and money received for childbirth are not really clear. I agree with Reviewer 1 the reference for each of these needs to be “No”, rather than “Don’t know”. It is possible to rearrange the data in the statistical package to do this. I also don’t understand why the emphasis is placed on being “somewhat satisfied” or “highly dissatisfied” with childbirth – I would have thought the hope was that the intervention increased the proportion of women who were positive about the experience, rather than emphasising dissatisfaction.  I also presume the asterisks indicate the level of significance. These are not needed, as the reader can see this from the p=values – the reader also needs to focus on the width of the confidence intervals, which are more meaningful than a p-value.

The supplementary table was not available for view in this revision. Please provide it again. 

We would appreciate receiving your revised manuscript by Jun 12 2020 11:59PM. To enhance the reproducibility of your results, we recommend that if applicable you deposit your laboratory protocols in protocols.io, where a protocol can be assigned its own identifier (DOI) such that it can be cited independently in the future. For instructions see: http://journals.plos.org/plosone/s/submission-guidelines#loc-laboratory-protocols

We look forward to receiving your revised manuscript.

Kind regards,

Christine E East

Academic Editor

PLOS ONE

---

## [Author Response · Author response to Decision Letter 1]

5 May 2020

Abstract

- Please add numeric results to the results section

Thank you for your comment, we have added numeric data to the results section of the abstract.

-Avoid stating “significantly more likely”. The post-intervention survey provided evidence that women were more likely to … than prior to the intervention.

Thank you for the suggestion. We have used following sentences in the results section of abstract “The post-intervention survey provided evidence that women were more likely to give birth at primary care facilities (OR 5.60, p-value <0.001) than prior to the intervention. Similarly, the likelihood of giving birth at a health facility increased if decision for birthplace was made jointly by women and family members for primary care facilities (OR 1.76, p-value 0.023) and hospitals/tertiary care facilities (OR 1.78, p-value 0.020). If women had less than four ANC visits, then they were less likely to give birth at primary care facilities (OR 0.39, p-value <0.001) or hospitals/tertiary care facilities (OR 0.63, p-value 0.014). Finally, women were less likely to give birth at primary care facilities if they had only primary level of education (OR 0.49, p-value 0.014)”

In the tracked copy, P5. Line 73. Need to start a sentence with a word, not an abbreviation.

We have done this.

Table 3. This needs the number and percent for each response. Some of these are already in Table 1. However, important factors such as knowledge about abortion and money received for childbirth are not really clear. I agree with Reviewer 1 the reference for each of these needs to be “No”, rather than “Don’t know”. It is possible to rearrange the data in the statistical package to do this. I also don’t understand why the emphasis is placed on being “somewhat satisfied” or “highly dissatisfied” with childbirth – I would have thought the hope was that the intervention increased the proportion of women who were positive about the experience, rather than emphasising dissatisfaction. I also presume the asterisks indicate the level of significance. These are not needed, as the reader can see this from the p=values – the reader also needs to focus on the width of the confidence intervals, which are more meaningful than a p-value.

Thank you for this important suggestion. As you have mentioned, we have not included the number and percent for each response in Table 3, since it is already mentioned in Table 1 and 2 and it would not be possible to include this Table 3 due to space constraints. Your comments about using ‘don’t know’ as reference category are valid, and we have rearranged the data in the SPSS for ‘knowledge about abortion’ and ‘money received for childbirth’. The reference category for these variables now are ‘No’ rather than ‘Don’t know’. We had to conduct the regression analysis again, therefore there is slight change in the figures in Table 3. The categories for ‘satisfaction with childbirth’ was based on the pre-intervention survey questionnaire and was not revised for the post-intervention survey. We understand that including this in the regression analysis does not clarify anything, so we chose not to include this in this analysis. 

As per your suggestion we have removed the asterisk sign as the confidence interval does tell this.

The supplementary table was not available for view in this revision. Please provide it again. 

We have provided this again.

---

## [Editor Report · Decision Letter 2]

11 May 2020

Evaluation of a health promotion intervention associated with birthing centres in rural Nepal

PONE-D-19-33046R2

Dear Dr. Mahato,

We are pleased to inform you that your manuscript has been judged scientifically suitable for publication and will be formally accepted for publication once it complies with all outstanding technical requirements.

With kind regards,

Christine E East

Academic Editor

PLOS ONE
---

## [Editor Report · Acceptance letter]

13 May 2020

PONE-D-19-33046R2 

Evaluation of a health promotion intervention associated with birthing centres in rural Nepal 

Dear Dr. Mahato:

I am pleased to inform you that your manuscript has been deemed suitable for publication in PLOS ONE. Congratulations! Your manuscript is now with our production department. 

With kind regards,

on behalf of

Dr. Christine E East 

Academic Editor

PLOS ONE